Avoiding “conflicts of interest”: a computational approach to scheduling parallel conference tracks and its human evaluation

Manda Prashanti p_manda@uncg.edu 1
Hahn Alexander 2
Beekman Katherine 3
Vision Todd J. 4
1 Department of Computer Science, University of North Carolina at Greensboro , Greensboro , NC , United States of America
2 Department of Computer Science, University of Southern California , Los Angeles , United States of America
3 ROI Revolution Inc. , Raleigh , NC , United States of America
4 Department of Biology, University of North Carolina at Chapel Hill , Chapel Hill , NC , United States of America
Mungall Christopher
Electronic publication date: 2019 Nov 11
Publication date: 2019
Volume: 5
Electronic Location ID: e234
Received 2019 Jan 21; Accepted 2019 Oct 15
Copyright: ©2019 Manda et al.
Copyright year: 2019
Copyright holder: Manda et al.
License: This is an open access article distributed under the terms of the Creative Commons Attribution License, which permits unrestricted use, distribution, reproduction and adaptation in any medium and for any purpose provided that it is properly attributed. For attribution, the original author(s), title, publication source (PeerJ Computer Science) and either DOI or URL of the article must be cited.
License URL: https://creativecommons.org/licenses/by/4.0/

Keywords: Topic modeling, Optimization, Conference scheduling

Funding: National Evolutionary Synthesis Center and the National Science Foundation EF-0905606 DBI-1062542 This work was supported by the National Evolutionary Synthesis Center and the National Science Foundation through EF-0905606 and DBI-1062542. The funders had no role in study design, data collection and analysis, decision to publish, or preparation of the manuscript.

==============================
Conferences with contributed talks grouped into multiple concurrent sessions pose an interesting scheduling problem. From an attendee’s perspective, choosing which talks to visit when there are many concurrent sessions is challenging since an individual may be interested in topics that are discussed in different sessions simultaneously. The frequency of topically similar talks in different concurrent sessions is, in fact, a common cause for complaint in post-conference surveys. Here, we introduce a practical solution to the conference scheduling problem by heuristic optimization of an objective function that weighs the occurrence of both topically similar talks in one session and topically different talks in concurrent sessions. Rather than clustering talks based on a limited number of preconceived topics, we employ a topic model to allow the topics to naturally emerge from the corpus of contributed talk titles and abstracts. We then measure the topical distance between all pairs of talks. Heuristic optimization of preliminary schedules seeks to balance the topical similarity of talks within a session and the dissimilarity between concurrent sessions. Using an ecology conference as a test case, we find that stochastic optimization dramatically improves the objective function relative to the schedule manually produced by the program committee. Approximate Integer Linear Programming can be used to provide a partially-optimized starting schedule, but the final value of the discrimination ratio (an objective function used to estimate coherence within a session and disparity between concurrent sessions) is surprisingly insensitive to the starting schedule. Furthermore, we show that, in contrast to the manual process, arbitrary scheduling constraints are straightforward to include. We applied our method to a second biology conference with over 1,000 contributed talks plus scheduling constraints. In a randomized experiment, biologists responded similarly to a machine-optimized schedule and a highly modified schedule produced by domain experts on the conference program committee.

Introduction

Researchers and educators depend upon professional conferences to showcase their work and stay current on the work of their peers. Thousands of such conferences are held each year worldwide, and conferences that feature of hundreds of oral presentations are not unusual. Such large conferences often schedule oral presentations in concurrent sessions so that each presentation can be allocated adequate time while keeping the overall conference duration to only a few days.

Conference scheduling is typically done manually by program organizers who review the large volume of talk submissions, decide which talks are similar to each other, and group similar talks into sessions accordingly (Fig. 1). They do this based on the information provided by prospective presenters, which invariably includes a title but may also include keywords, topic categories and/or an abstract. This is a tedious and often error-prone process, done in some cases under considerable time pressure, that is not easily scaled and can lead to sub-optimal conference schedules (Hillis, 2013). Since conference attendees typically aim to attend those talks most relevant to their interests, the ideal conference schedule will not only ensure similarity of topics within a session, but also avoid topical conflict among concurrent sessions.

Figure 1 Two steps in the process of manually assigning talks to sessions for the 2017 American Physical Society March Meeting.

Photos courtesy of Dr. Karen Daniels. Photo credit to Dr. Daphne Klotsa.

In practice, identifying similarity among talks is a highly subjective process. Research talks often have several dimensions; a talk presenting an efficient key distribution scheme for asymmetric cryptography is related to key distribution algorithms, network security, and cryptographic algorithms. Talk A might be more similar to talk B on one dimension but more similar to talk C on a different dimension. Depending on their areas of expertise, different organizers might weight those dimensions differently, and the weights of the organizers may or may not be representative of the conference attendees.

Even if the measure of similarity were not subjective, ensuring a high level of dissimilarity among concurrent sessions, each with multiple talks, is a challenging task for humans, as it requires perception of the distribution of many points in a highly multidimensional space. This can lead to schedules with conflict between concurrent sessions even when the talks within each individual session appear similar. Ensuring a high level of dissimilarity among concurrent sessions is important to minimize participants having to move between sessions, or having to choose between co-occurring talks of equal interest. Vangerven et al. (2018) also note that dissimilarity between concurrent sessions is important for enabling participants to attend the talks of most interest to them without encountering scheduling conflicts, as might happen when talks of a similar topical nature are scheduled in concurrent sessions.

Adding to the complexity of the conference scheduling task is the fact that organizers typically have to accommodate idiosyncratic scheduling constraints due to the travel schedules and other obligations of individual presenters. Efficient and automated data-driven solutions to overcome the problems would be desirable.

The conference scheduling problem

Imagine a conference with Q talks scheduled across W days with a maximum of N timeslots per day, each with a maximum of Cmax concurrent sessions. A session is defined as a sequence of talks scheduled during one timeslot in one room. The maximum number of talks in a session is predefined by the organizers and does not vary across the schedule. Sessions are considered concurrent when they are scheduled in the same timeslot. Timeslots are non-overlapping.

We define the conference scheduling problem as the task of assigning talks to timeslots and concurrent sessions so as to maximize coherence within a session and minimize similarity between concurrent sessions (i.e., those within the same timeslot). In this work, we describe a heuristic solution to the conference scheduling problem that creates optimized conference schedules with multiple concurrent sessions in a fully automated fashion.

First, we propose the use of a data-driven machine-learning approach, topic modeling (Wallach, 2006), to infer similarity between talks. We use topic modeling to identify a set of latent topics relevant to the full set of talks being presented at a conference. Each talk can then be represented as a weighted vector of these different topics, and we can compare these vectors as a measure of similarity. Thus, topic modeling provides a principled way to decide upon which dimensions to consider, and how to weigh those dimensions, in measuring similarity (between talks, between sessions, or between talks and sessions).

Second, we present a suite of heuristic schedule creation approaches designed to maximize an objective function that quantifies session coherence and dissimilarity between concurrent sessions in a single metric. We explore different strategies to create initial schedules, including a greedy heuristic, random assignment, and Integer Linear Programming. We then explore different stochastic optimization strategies to further improve upon the initial schedules (Spall, 2012), and investigate how the optimality of the initial schedule impacts the final result.

We selected a high-performing combination of approaches that improved upon a manually produced schedule for a recently held ecology conference. Using this combination of approaches, we then created an automated schedule for a large evolutionary biology conference that had not yet been held, in collaboration with the conference organizing committee. The organizing committee made major, manual modifications to produce the final schedule that was used.

After the evolutionary biology conference was held, we conducted an experiment where biologists with expertise in that field were presented with samples of the concurrent sessions from both the machine-generated and manually-modified schedules in order to elicit their subjective opinions about session coherence and conflict among concurrent sessions. This provided an evaluation of how well the discrimination ratio captured the topic dimensions that mattered to experts in the field who would be representative of conference attendees.

Related work

Surprisingly, given the pervasive exposure of academics to the challenges of conference scheduling, there is a relatively small literature on the problem (reviewed in Vangerven et al. (2018)). Bhardwaj et al. (2014) and André et al. (2013) incorporate attendee preferences for talk and session placement in a Community-Informed Conference Scheduling approach (Sampson, 2009; Kim et al., 2013; Chilton et al., 2014). In Sampson (2009), conference attendees submit time preferences for their talk. The scheduling algorithm, a modification of the simulated annealing heuristic, then attempts to accommodate participant preferences using the number of preferences accommodated as the objective function. Similarly, Kim et al. (2013) and Chilton et al. (2014) use community sourced preferences for talk and session placement to guide the scheduling process. Conference attendees are asked to submit their preferences as to which talks should be scheduled with their own talk and which talks belonged in similarly themed sessions that should not be scheduled concurrently to one another. These preferences are encoded into a scheduling interface that is then used by organizers to create and schedule sessions with the aim of maximizing the number of preferences accommodated while resolving author, talk, and session conflicts. In contrast to our approach, the actual scheduling process is manual.

Edis & Edis (2013) approach the scheduling problem using an Integer Programming (IP) formulation in which each talk is assigned a topic and talks are assigned to a session so that all talks in the session have the same topic. Houlding & Haslett (2013) describe a clustering algorithm to group similar talks into fixed size clusters (or sessions), using a local objective function that maximizes the similarity of talk pairs assigned to a cluster at each step. To measure similarity between talks, participants are asked to select three relevant sessions for their talk. The co-occurrence frequency of session topics is then used to determine similarity between talks and sessions. Gulati & Sengupta (2004) use an augmented objective function that incorporates a prediction of a talk’s popularity based on reviewer comments and participant preferences of time slots. The goal of the schedule is to maximize session attendance. The work uses a greedy scheduling algorithm, but no empirical results or computational analysis are presented. Ibrahim, Ramli & Hassan (2008) focus on assigning talks to time slots across a number of days in 3 concurrent sessions. Each talk belongs to a field or topic and the goal is avoid scheduling talks of the same topic concurrently. The study presents methods based on combinatorial design theory for three conferences used as case-studies. The study does not address how talks are grouped into sessions. Quesnelle & Steffy (2015) consider an optimization problem that assigns talks to timeslots and rooms such that scheduling conflicts are minimized while accounting for presenter and room availabilities.

Potthoff & Munger (2003) apply Integer Programming to assign sessions to time periods in a way that sessions for each subject area are spread evenly across time slots. Similarly, Nicholls (2007) assign sessions to rooms and time periods to avoid presenters being scheduled in two concurrent sessions while trying to maximize presenter preferences in the schedule. Both of the above assume that the clustering of similar talks into sessions has already been accomplished. Nicholls (2007) and Eglese & Rand (1987) aim to optimize participant satisfaction by collecting participant preferred sessions. In Eglese & Rand (1987), a simulated annealing algorithm assigns sessions to time periods with the aim of minimizing the sum of the weighted violations of session preferences. Le Page (1996) requires participants to provide the number of sessions they would like to attend. They build a conflict matrix containing the number of people that wish to attend both session i and j. The goal is to assign sessions to timeslots such that the sum of conflicts between simultaneous sessions is minimized. Sessions with the same topic must be assigned to the same room. The authors propose a semi-automated heuristic consisting of four steps that was used to schedule a meeting of the American Crystallographic Association.

Among the few studies to address the problem of grouping similar talks into sessions were Tanaka, Mori & Bargiela (2002) and Tanaka & Mori (2002). They use a set of user assigned keywords for each talk and use an objective function that is a nonlinear utility function of common keywords. The intuition behind the approach is that papers in the same session have as much overlap in keywords as possible. They use Kohonen’s self organizing maps (Tanaka, Mori & Bargiela, 2002) and a hybrid grouping genetic algorithm (Tanaka & Mori, 2002). Vangerven et al. (2018) present a method that approaches the conference scheduling problem in three phases. The first phase aims to maximize total attendance, based on the participants’ preferences. The second phase tries to minimize the total number of session hops or minimizing the amount of topical overlap between concurrent sessions. The third phase aims to accommodate presenter preferences and availabilities by minimizing the total number of preferences violated.

Stidsen, Pisinger & Vigo (2018) approach conference scheduling using a number of optimization models each with a specific objective. Research fields are assigned to buildings with the aim of assigning related areas to buildings physically close to each other. Each session is assigned to one room. Finally, the solution optimizes assignment of sessions to room sizes.

Despite these research contributions, the practice of manual scheduling is still widespread, and not all factors that would allow for a practical automated solution have been considered by researchers.

Compared to previous work, our approach is novel in its use of topic modeling to measure talk similarity in multiple dimensions, stochastic optimization of a global objective function that ensures both similarity within a session and disparity between concurrent sessions, and the lack of a need for human intervention.

Methods

We first provide a description of the different parameters and variables (‘Preliminaries’) used through ‘Methods’. Then, we present details about creating the corpus of documents (‘Creating the corpus for topic modeling) for topic modeling along with topic modeling algorithms used (‘Topic modeling’). Next, we describe how similarity between talks and sessions will be computed using outputs from the topic model (‘Computing similarity between talks and sessions). An objective function called the Discrimination Ratio to quantify the similarity of talks within a session vs. disparity between concurrent sessions will be presented (‘An objective function for conference scheduling ’An objective function for conference scheduling’). Finally, we outline heuristic approaches for creating initial schedules (‘Creation of initial schedules’) and for optimizing the initial schedules (‘Stochastic optimization’).

Preliminaries

A conference schedule is composed of W days, each with N timeslots, with a total of Q talks. Each timeslot is further divided into a maximum of Cmax concurrent sessions. Two sessions are considered to be concurrent if the starting and ending time of the sessions are the same. The number of concurrent sessions in any given timeslot i is represented by Ci. Each session can contain a maximum of Tmax talks. A session is a sequence of talks scheduled during one timeslot in one room. For a given session j, the number of talks in the session is represented by Tj. Talks in a particular timeslot and a particular session can be referred to in the order in which they’re scheduled. ti,j,k represents the kth talk in session j of timeslot i.

The topic modeling algorithm takes as input the number of topics (G) to be generated from the corpus of Q talks. The algorithm outputs a vector representation (Vi →) of each talk as a weighted vector over the G topics. The vector contains the probabilistic relevance of each topic to a talk. For example, Vi,1 is the probabilistic relevance of topic 1 to talk ti (the ith talk in a session). A pairwise similarity matrix (M) is computed from the above vector representation that contains the cosine similarity (S) between vectors (V1 →,V2 →) of every pair of talks in the corpus. The cosine similarity, S, of two vectors has a minimum of -1 and a maximum of 1. An objective function, Discrimination Ratio (D), is defined as the ratio of the mean talk similarity within a session (Sw) to the mean talk similarity between concurrent sessions (Sb) across the full schedule.

Initial schedules can be created using the Random, Greedy, or ILP approaches (‘Creation of initial schedules’). These approaches take the number of days in the conference (W), number of timeslots per day (N), maximum number of concurrent sessions in a timeslot (Cmax), maximum number of talks in each concurrent session (Tmax), and the pairwise talk similarity matrix (M).

We present two variants each of a Hill Climbing (HC) and a Simulated Annealing (SA) algorithm that further optimize the initial schedules. For the HC and SA approaches, we experiment with a version (HCA, SAA) that optimizes the objective function directly and another (HCA, SAA) that splits the optimization into two stages - first maximizing within-session similarity and then minimizing between-session similarity. All four variants (HCA, HCB, SAA, SAB) take a starting schedule (I), the number of parallel optimization runs (R), the maximum number of swaps (e), and a pairwise talk similarity matrix (M). The approaches can optionally take a list of scheduling constraints encoded as a dictionary (L). In addition, the SA versions (SAA SAB) take an initial temperature (Z) and a constant (α = 0.99). These parameters are further defined in ‘Simulated annealing’.

For ease of reference, the parameters and variables are listed in Table 1.

Table 1 Parameters and variables used in the topic modeling, schedule creation, and optimization approaches.

Parameter	Description	
I	Input starting schedule created using the Random, Greedy, or Integer Linear Programming (ILP) approaches.	
D	Discrimination ratio	
N	Number of timeslots in a schedule	
Ni	Timeslot i	
Cmax	Maximum number of concurrent sessions in a timeslot	
C	Number of concurrent sessions	
Ci	Number of concurrent sessions in timeslot i	
Tmax	Maximum number of talks in a session	
T	Number of talks	
Tj	Number of talks in session j	
Q	Number of talks in a schedule	
ti,j,k	kth talk in session j of timeslot i	
SV1 →,V2 →	Cosine similarity between the vector representations of two talks	
Sw	Mean intra-session similarity of a schedule	
Sb	Mean inter-session similarity of a schedule	
M	Pairwise talk similarity matrix	
Y	Number of seed talks for the Greedy algorithm	
X	Number of clusters created by the Kruskal’s algorithm	
Xi	ith Kruskal’s cluster	
TXi	Number of talks in cluster Xi	
AXi	Attractor talk for cluster i	
Z	Initial temperature	
Zi	Temperature at the ith swap. Z0 = 50, 000.	
α	Constant set to 0.99	
R	Number of parallel optimizations runs for an optimization algorithm	
e	Number of maximum swaps for an optimization algorithm	
W	Number of days in a conference	
L	Dictionary of scheduling constraints	
G	Number of topics in the topic model	
Vi,j	Probabilistic relevance of topic j to talk ti	

Creating the corpus for topic modeling

The corpus of documents that is input to the topic model is the set of talks for a conference. In our implementation, each document included the title and abstract for a single talk. To ensure that the corpus only contained meaningful words that reflect the semantic content of talks, stemming and stop word removal were applied. Stemming reduces variants of words to their base or root form (Lovins, 1968; Porter, 2001; Porter, 1980), making it easier for the topic modeling algorithm to recognize words with the same meaning. Stop words are commonly used words (such as ‘and’, ‘it’, and ‘the’) that have little value with respect to the meaning of the text (Fox, 1989). Python’s Natural Language Toolkit (NLTK - https://www.nltk.org) provides a set of 179 commonly used english words that was used as the initial stop word list.

For the second conference, Evolution 2014, domain experts on the conference organizing committee added additional stop words, leading to a total of 952 stop words.

Topic modeling

We used Latent Dirichlet Allocation (LDA), a generative probabilistic model often used to describe collections of text corpora and one of the most widely used topic modeling algorithms (Blei, Ng & Jordan, 2003). LDA models each document as a finite mixture over an underlying set of latent topics, and each latent topic as a probabilistic mixture over relevant words. The model assumes Dirichlet priors over the latent topics in a document and relevant words within a topic.

One of the input parameters to the LDA algorithm is the number of topics to identify from the corpus. Several preliminary topic models were created using different numbers. We developed a metric, the Match Percentage, to compare the fit of different models. For each model, the top two words from each of the top three topics of a talk were used to create a set of six keywords. The fraction of keywords found within the title and abstract was computed for each talk and the Match Percentage was computed as the mean of this fraction across all talks, expressed as a percentage. The topic model with the highest Match Percentage was chosen for subsequent analyses.

While there are automated metrics, such as perplexity (Blei & Lafferty, 2005), to evaluate topic models, studies that have tested these metrics of evaluating topic models have reported that inferences based on these measures were negatively correlated with human perception (Chang et al., 2009; Chang & Blei, 2009). These studies also suggest that topic models should be chosen by human analysis of coherence of topics inferred by a model, words in topics etc. instead of trying to optimize likelihood based measures (Chang & Blei, 2009).

Computing similarity between talks and sessions

LDA outputs a representation of each talk in the corpus as a weighted vector over all the latent topics. In a model with G topics, the vector Vi → of talk ti is defined as (1) Vi →=Vi,1,Vi,2,…,Vi,G

where iis the talk number

Vi,1is the probabilistic relevance of topic 1 to talkti

From this, a pairwise similarity matrix, M, is computed by calculating the cosine similarity (S) of the two vectors, V1 → and V2 →, for every pair of talks in the corpus. (2) SV1 →,V2 →=∑j=1GV1,jV2,j∑j=1GV1,j2∑j=1GV2,j2.

An objective function for conference scheduling

We introduce an objective function called the Discrimination Ratio, D, to quantify in one measure the similarity of talks within each session and the disparity between talks in concurrent sessions. D is defined as the ratio of the mean within-session similarity to the mean between-session similarity across the full schedule. D is higher (>1) when the mean within-session similarity is higher as compared to mean between-session similarity in a schedule. Lower D values (<1) indicate that the mean within-session similarity is lower as compared to mean between-session similarity in a schedule. D is 1 when the mean within-session similarity is same as the mean between-session similarity.

The mean within-session similarity, Sw, is the mean of the pairwise similarities between all the talks within each session. (3) Sw=∑i=1N ∑j=1Ci ∑k=1Tj−1 ∑l=k+1TjSti,j,k,ti,j,l∑i=1N ∑j=1CiTj2

where N is the number of timeslots in the schedule, Ci is number of concurrent sessions in timeslot i, Tj is number of talks in session j, and S(ti,j,k, ti,j,l) (from Eq. (3)) is the cosine similarity between talk k in timeslot i, session j and talk l in timeslot i, session j.

The mean between-session similarity, Sb, is the mean of the pairwise similarities between all the talks in different concurrent sessions. (4) Sb=∑i=1N ∑j=1Ci ∑k=1Tj ∑l=j+1Ci ∑m=1TlSti,j,k,ti,l,m∑i=1N ∑j=1Ci ∑k=1Tj ∑l=j+1Ci ∑m=1Tl1

The Discrimination Ratio is defined as D = Sw∕Sb. D is inspired by other commonly used metrics used to evaluate the quality of clusters generated by clustering algorithms, such as k-means. Such commonly used metrics include the Error Sum of Squares (SSE)—the sum of the squared differences between each observation and its cluster’s mean (Celebi, Kingravi & Vela, 2013), Intercluster Distance (Gonzalez, 1985)—the sum of the squared distances between each cluster’s centroid, or Intracluster Distance—the sum of the squared distances between an item and its cluster’s centroid.

Creation of initial schedules

We consider three approaches for the creation of initial schedules: random, Greedy, and Integer Linear Programming (ILP).

Random

The Random assignment algorithm provides a baseline against which to compare the performance of approaches that explicitly optimize the objective function. Given a set of talks and scheduling parameters as in ‘Preliminaries’, this algorithm assigns talks to sessions through sampling with replacement with no consideration of talk similarities or the value of the objective function.

Greedy

The Greedy assignment algorithm generates a semi-optimal schedule for further stochastic optimization. In addition to the parameters in ‘Preliminaries’, the algorithm requires a set of Y seed talks that are selected based on an input threshold of minimum dissimilarity between each other. First, the algorithm finds a session for each seed talk such as to maximize the objective function. Next, the rest of the talks are assigned to sessions by choosing the most locally optimal solution at each step.

Integer linear programming

We cast the problem of scheduling the conference as an Integer Linear Program (ILP) using a variable reduction technique that was solved using AMPL (Gay & Kernighan, 2002) with the CPLEX solver (http://www.cplex.com).

An Integer Linear Program (ILP) consists of variables, constraints, and an objective function where some or all of the variables take on integer values (Bosch & Trick, 2005). Non-integer variables have numeric values that are limited to a feasible region by the constraints. The objective function determines the assignment of values to the variables that results in an optimal solution. Both the constraints and the objective function must be linear in the variables.

In our implementation, a heuristic pre-processing step first groups the talks into X clusters of similar talks using a modified version of Kruskal’s algorithm (Kruskal, 1956), a greedy algorithm that is used to find a minimum spanning tree from a weighted graph of nodes. In this work, nodes represent talks while edge weights represent pairwise talk similarity. We use a modification of Kruskal’s algorithm to find a number of disjoint maximum-weight spanning trees from the graph. Each disjoint spanning tree is a cluster that groups similar talks while the spanning trees are sufficiently distant from each other. At the beginning of the algorithm, each talk forms its own cluster. At each iteration of the algorithm, the pair of talks with the highest edge weight (similarity score) is selected. If the two talks are in separate clusters, the clusters are merged to form a bigger cluster. The algorithm is terminated as soon as X disjoint and distant clusters of similar talks are created.

A representative talk called the attractor (AXi), is then chosen from each of the X clusters. The aim is to produce a set of initial input talks for the ILP that are highly different from each other, while ensuring that each attractor has many other talks similar to it. We choose as the attractor the talk that has the highest similarity to all other talks in its cluster. If there are multiple talks that qualify as attractors, one of those talks is chosen randomly. We calculate a fit score (F) for each talk tj in cluster Xi as follows. (5) Ftj,Xi= maxk=1TXiStj,tk:j≠k

where TXiis the number of talks in clusterXi

The talk tj with the maximum value of F is chosen as the attractor for cluster Xi.

This list of attractors is then input to the ILP, which optimally assigns one attractor to each concurrent session in the schedule and assigns talks to sessions so as to maximize the sum of similarities between the attractor and all the other talks in that session.

In addition, the ILP requires the following constraints: each session i is assigned no more than Tmax talks, exactly Ci attractors must be assigned to each timeslot Ni, and each talk must be assigned to only one session.

We made no effort to ensure distinctness of the initial schedules either within or between the three approaches.

Stochastic optimization

We developed two variants each of a Hill Climbing algorithm and a Simulated Annealing algorithm to further improve upon the initial schedules (obtained from the Random, Greedy, and ILP approaches) by iteratively proposing swaps in the positions of talks in the schedule. The Hill Climbing (HC) approaches accept solutions from a swap only when they increase the discrimination ratio, and are thus susceptible to being trapped in local optima. By contrast, the simulated annealing (SA) approaches will accept solutions that decrease D with a certain probability, and thus have the potential/possibility to escape local optima (Kirkpatrick, Gelatt & Vecchi, 1983).

Each optimization algorithm takes one or more initial schedules as input and spawns R parallel optimization runs to produce R optimized schedules at the end of execution. If the input schedule is Random, each parallel run starts with an independently generated schedule, while if the input schedule is a Greedy or an ILP schedule, all parallel runs operate on the same input. The schedule with the highest discrimination ratio among the R optimized schedules is chosen as the output of the algorithm.

The input parameters to the optimization approaches are given in ‘Preliminaries’.

Simulated annealing

For simulated annealing, we used the Kirkpatrick acceptance probability function (Eq. (6)) to determine the probability of accepting a solution resulting from a swap (Kirkpatrick, Gelatt & Vecchi, 1983). (6) KDj,Di=1ifDj<Dire−Dj−Di∕Ziotherwise

where Di and Dj are the discrimination ratios of the schedule under the proposed swap and after the last accepted swap, respectively; Z is the initial “temperature”, and Zi is the current temperature at timestep i defined as Zi = Zi−1α.

The decreasing temperature values reduce the probability of accepting worse solutions as the number of swaps increases. Since the algorithm might accept worse solutions, the best solution encountered at any point of time is stored to be reported at the end of execution.

Sequential optimization

In working with the organizing committee for the evolution conference, we observed that users were more sensitive to maximizing coherence within a session than disparity between concurrent sessions. In order to emulate this aspect of human scheduling, we developed variants of the HC and SA approaches that split the optimization algorithm into two sequential regimes, the first optimizing for within-session similarity alone and the second for between-session disparity alone. Between-session disparity is optimized by proposing a swap of two randomly selected sessions in each iteration. This has no effect on within-session similarity since swapping is conducted on sessions and not talks. The sequential optimization regimes are stopped when further swapping does not result in improvement.

We refer to the versions of the HC and SA approaches in which D is optimized directly throughout as HCA and SAA, respectively, and the approaches in which the schedule is first optimized for within-session similarity as HCB and SAB, respectively. See Algorithms 1–4 for pseudocode describing the four optimization approaches.

Scheduling constraints

In practice, a conference will typically have constraints that restrict the sessions or timeslots a talk can be placed in. Reasons may include talks competing for awards that must scheduled early in the conference in order to allow time for judging; presenters with multiple talks that cannot be scheduled in concurrent sessions within the same timeslot; presenters who are scheduled to arrive at the conference after it begins or before it is finished; or requests for complementary talks to be scheduled in the same session.

These constraints can be accommodated by the optimization approaches described above by requiring them to be satisfied in any solution obtained. In our implementation, such scheduling constraints were encoded as a dictionary (L) that maps each talk to a set of sessions in which the talk can be placed without violating any scheduling constraints. For example, in a schedule with five sessions (labeled 1 through 5), if a constraint prevents talk ti from being scheduled in session 5, the constraint would be encoded in the dictionary as L[ti]={1,2,3,4}. Each proposed swap was checked for constraint violations before being accepted. If there is no feasible solution due to conflicting constraints, no solution is returned.

Results

The datasets for the two conferences used in this work, Ecology 2013 and Evolution 2014, are summarized in Table 2. We first tested our topic modeling, schedule creation and optimization approaches on select concurrent sessions from Ecology 2013. The manually created schedule for this conference gave us a point of comparison for the automated schedules we generated. Although our ultimate goal was to apply our methods to the Evolution 2014 conference, previous Evolution conferences could not be used for testing since talk abstracts were not a part of submissions in previous years. The main structural difference between the two datasets is that no scheduling constraints were available for Ecology 2013.

Table 2 Parameters for the Ecology 2013 and Evolution 2014 conferences.

Parameter	Ecology 2013	Evolution 2014	
Number of days (W)	5	4	
Number of talks to be scheduled (Q)	324	1,014	
Number of timeslots (N)	8	16	
Maximum number of concurrent sessions per timeslot (Cmax)	5	14 (W1, W2, W3), 9 (W4)	
Maximum number of talks per session (Tmax)	10	5	
Scheduling constraints to be accommodated	None	244	

____________________________________________________________________________  Algorithm 1: Hill Climbing optimization algorithm (HCA) ____________________________________________________________________________   Input:   Initial schedule   Output:   Optimized schedule   set current schedule to input schedule;   set e to maximum number of swaps;   while discrimination ratio (D) increases or e > 0  do     select two talks from the current schedule at random;     swap the two talks;     if  updated schedule does not violate constraints  then          compute D of updated schedule;          if  updated D > current D then               accept changes;               set current schedule to updated schedule;          end     else         discard changes;     end     e = e − 1;   end   return current schedule; ____________________________________________________________________________

________________________________________________________________________________  Algorithm 2: Hill Climbing optimization algorithm (HCB) ________________________________________________________________________________    Input:   Initial schedule    Output:   Intra-session optimized schedule    set current schedule to input schedule;    set e to maximum number of swaps;    while mean intra-session similarity (Sw) increases or e > 0  do        select two talks from the current schedule at random;        swap the two talks;        if  updated schedule does not violate constraints  then             compute Sw  of updated schedule;             if  updated Sw > current Sw  then                  accept changes;                  set current schedule to updated schedule;             end         else             discard changes;         end         e = e − 1;    end    set Intra-session optimized schedule to current schedule;    Input:   Intra-session optimized schedule    Output:   Optimized schedule    set e to maximum number of swaps;    set current schedule to input schedule;    while mean inter-session similarity (Sb) decreases or e > 0  do         select two sessions from the input schedule at random;         swap the two sessions;         if  updated schedule does not violate constraints  then              compute Sb  of updated schedule;              if  updated Sb < current Sb  then                   accept changes;                   set current schedule to updated schedule;              end         else              discard changes;         end         e = e − 1;    end    return current schedule ; ________________________________________________________________________________

Ecology 2013

Topic models were created using the LDA algorithm for 60, 80, 100, and 120 topics on the corpus of 324 talks. Each topic model was evaluated based on two criteria: (1) Match Percentage and (2) manual examination of the topics and topic words associated with each talk. We obtained Match Percentages of 70.5% (for 60 topics), 74.6% (80), 75.2% (100) and 75% (120). The topic model with 100 topics was judged to be the best model for the data. Subsequently, this topic model was used to compute a talk similarity matrix that contained a similarity score for all pairs of talks in the dataset. The talk similarity matrix was computed using cosine similarity between the topic relevance vectors of any two talks (Eq. (2)).

____________________________________________________________________________________  Algorithm 3: Simulated Annealing optimization algorithm (SAA) ____________________________________________________________________________________   Input:   Initial schedule   Output:   Optimized schedule   set e to maximum number of swaps;   set best D to D of input schedule;   set current schedule, best schedule to input schedule;   while discrimination ratio (D) increases or e > 0  do     select two talks from the input schedule at random;     swap the two talks;     if  updated schedule does not violate constraints  then          compute D of updated schedule;          compute probability of accepting updated schedule using Kirkpatrick accep          tance probability function;         r=random number between 0 and 1;         if  acceptance probability > r  then               accept changes;               set current schedule to updated schedule;               compute D of updated schedule;               if  updated D > current D then                    set best schedule to updated schedule               end         else            discard changes;         end     else        discard changes;     end     e = e − 1;   end   return best schedule; ______________________________________________________________________________________

______________________________________________________________________________________________________  Algorithm 4: Simulated Annealing optimization algorithm (SAB)  _____________________________________________________________________________________________________   Input: Initial schedule   Output: Optimized schedule   set e to maximum number of swaps;   set best Sw to Sw of input schedule;   set current schedule, best schedule to input schedule;   while mean inter-session similarity (Sw) increases or e > 0 do       select two talks from the current schedule at random;       swap the two talks;       if updated schedule does not violate constraints then            compute Sw of updated schedule;            compute probability of accepting updated schedule using Kirkpatrick acceptance probability             function;            r=random number between 0 and 1;            if acceptance probability > r then                 accept changes;                 set current schedule to updated schedule;                 if updated Sw > best Sw then                     best Sw = updated Sw;                     best schedule = updated schedule;                 end            else                discard changes;            end        else            discard changes;        end        e = e − 1;   end   set Intra-session optimized schedule = best schedule;   Input: Intra-session optimized schedule   Output: Optimized schedule   set e to maximum number of swaps;   set current schedule, best schedule to input schedule;   set best Sb to Sb of input schedule;   while mean inter-session similarity (Sb) decreases or e > 0 do       select two talks from the current schedule at random;       swap the two talks;       if updated schedule does not violate constraints then           compute Sb of updated schedule;           compute probability of accepting updated schedule using Kirkpatrick acceptance probability            function;           r=1-(random number between 0 and 1);           if acceptance probability > r then                accept changes;                set current schedule to updated schedule;                if updated Sb < best Sb then                    best Sb = updated Sb;                    best schedule = updated schedule;                end           else               discard changes;           end       else           discard changes;       end       e = e − 1;   end   return best schedule;   _________________________________________________________________________________________________________

Fifty each of Random, Greedy, and ILP schedules were created in addition to the manually created Ecology 2013 schedule. The schedule with the highest discrimination ratio among the 50 runs was taken to be the solution for each combination of starting schedule and stochastic optimization algorithm.

The discrimination ratios of the initial and final optimized schedules are shown in Fig. 2. Both the Greedy and ILP initial schedules outperformed the Manual schedule while the Random schedule did not. All four optimization approaches improved upon the initial schedules. The highest relative improvement was seen on the Random schedules (about eight-fold) while a two-fold improvement was seen relative to the other three initial schedules, yet the final schedules had very similar discrimination ratios irrespective of the initial schedule. Among the optimization approaches, the overall best results were obtained with SAA, closely followed by HCA, on all initial schedules. Thus, the two approaches optimizing directly and continuously for D outperformed those that sequentially optimized for within-session similarity followed by between-session disparity.

Figure 2 Mean discrimination ratio of the starting and final Ecology 2013 schedules for the four optimization approaches applied to each of the Random, Manual, Greedy, and ILP initial schedules.

Error bars show two standard errors of the mean discrimination ratio among the 50 starting or final schedules.

We compared the D distributions using a Student’s t-test across 50 final schedules created HC and SA approaches with different initial schedules to investigate if there are any significant differences in performances. We found statistically significant differences between SA and HC versions for the majority of starting schedules at the Bonferroni-corrected threshold of α = 0.002 (Table 3, rows 2–9). No significant differences were found between HCA and SAA with a Random starting schedule and between HCB and SAB with an ILP starting schedule (Table 3, rows 2,9).

Table 3 Comparison of HC and SA approaches, and, A and B variants of HC and SA with four different initial schedules.

Comparisons are made on the distributions of D for the 50 final Ecology 2013 optimized schedules produced by each approach. Shown are p-values from two-sided un-paired t-tests at the Bonferroni-corrected threshold of α = 0.002 (experiment-wide α = 0.05 for n = 24).

Initial Schedule	Comparison	p	
Comparing HC with SA versions			
Random	HCA vs. SAA	0.32	
Manual	HCA vs. SAA	4.56e−07	
Greedy	HCA vs. SAA	1.01e−05	
ILP	HCA vs. SAA	0.0002	
Random	HCB vs. SAB	7.28e−43	
Manual	HCB vs. SAB	2.77e−50	
Greedy	HCB vs. SAB	6.12e−38	
ILP	HCB vs. SAB	0.084	
Comparing A with B variants			
Random	HCA vs. HCB	9.03e−55	
Manual	HCA vs. HCB	2.09e−58	
Greedy	HCA vs. HCB	1.30e−52	
ILP	HCA vs. HCB	2.36e−54	
Random	SAA vs. SAB	4.18e−23	
Manual	SAA vs. SAB	4.07e−23	
Greedy	SAA vs. SAB	5.90e−23	
ILP	SAA vs. SAB	2.36e−54	

We also compared the performance of A and B versions of the HC and SA approaches with the four initial schedules. Statistically significant differences were found between A and B versions for both HC and SA approaches across all four starting schedules (Table 3, rows 11–18).

Evolution 2014

Topic models were created from the corpus of 1,014 talks using the LDA algorithm for 50, 100, 150, and 250 topics. We obtained Match Percentages of 72.4% (for 50 topics), 76.8% (100), 79.3% (150) and 77.2% (250). Based on the match percentage of the four models and manual inspection of the generated topics, the model with 150 topics was chosen to compute talk similarity for the Evolution 2014 corpus.

During the test runs conducted on the Ecology dataset, we observed that there was little variation between different parallel runs within the same algorithm (Fig. 2). Knowing this, and considering the larger size of the Evolution 2014 dataset, we reduced the number of parallel runs for each optimization algorithm to 10. Since the Ecology 2013 results showed that the initial schedule had no discernible affect on the final optimized schedule, we only report the results of optimization on Random starting schedules with and without constraints.

The results are shown in Fig. 3. The relative ordering of the approaches is identical to Ecology 2013, with the highest performance shown by SAA followed closely by HCA. Interestingly, the inclusion of constraints did not lead to a reduction in the discrimination ratios; in fact, the highest discrimination ratio (6.7) was obtained in the presence of constraints.

Figure 3 Mean discrimination ratio of starting and optimized Evolution 2014 schedules for the four optimization approaches applied to Random initial schedules with and without constraints.

Error bars show two standard errors of the mean discrimination ratio among the 10 starting or final schedules.

We compared the D distributions using a Student’s t-test across 10 final schedules created HC and SA approaches with a Random initial schedule. Statistically significant differences were found between SA and HC versions with a Random initial schedule both with and without additional scheduling constraints (Table 4, rows 2–5).

Table 4 Comparison of HC and SA approaches, and, A and B variants of HC and SA with four different initial schedules.

Comparisons are made on the distributions of D for 10 final Evolution optimized schedules produced by each approach. Shown are p-values from two-sided un-paired t-tests at the Bonferroni-corrected threshold of α = 0.002 (experiment-wide alpha = 0.05 with n = 24).

Initial Schedule	Comparison	p	
Comparing HC with SA versions			
Random	HCA vs. SAA	1.95e−08	
Random	HCB vs. SAB	8.44e−35	
Random with Constraints	HCA vs. SAA	1.54e−08	
Random with Constraints	HCB vs. SAB	4.06e−23	
Comparing A with B variants			
Random	HCA vs. HCB	3.23e−41	
Random with Constraints	HCA vs. HCB	8.52e−27	
Random	SAA vs. SAB	1.29e−21	
Random with Constraints	SAA vs. SAB	9.44e−23	

We also compared the performance of A and B variants of the HC and SA approaches. Statistically significant differences were found between A and B versions for both HC and SA approaches both with and without scheduling constraints (Table 4, rows 7–10).

Preliminary labels can be generated for the automated sessions using information from the topic model. For example, for each talk in a session, we can determine the top two words from the top two relevant topics that describe the talk the best. This would result in a set of four words (assuming no redundancies) that represent each talk. The most frequently occurring words among the talks can be used to create a preliminary label, which can then be used to construct a session name by program organizers.

Manual modification of Evolution 2014 schedule

The SAA schedule with constraints, reported above, was then given to the Evolution 2014 program committee as a starting point. The program committee consisted of ten evolutionary biologists. Based on their subjective judgments, and following manual procedures that elude easy description, the committee members made a large number of changes to the placement of talks and sessions before finalizing the schedule for the conference. In addition, the program committee added extra sessions for special symposia that were not part of the pool of contributed talks.

The changes made by the program committee were substantial; 0.50% of talk pairs that shared a session in the automated schedule were retained within the same session in the modified schedule, while 4.40% of talk pairs placed in concurrent sessions in the modified schedule had originally been placed together in the automated schedule. The value of D for the original automated schedule was 6.7, while that for the manually modified schedule was 3.2.

Expert evaluation

The differences between the automated and manually modified Evolution 2014 schedule provided an opportunity to conduct a human evaluation. We were particularly interested in comparing how tempted users would be to hop between sessions in each case. To that end, we presented a set of 29 volunteers with expertise in evolutionary biology, none of whom served on the program committee, with faux schedules compiled from the two different sources. Responses were captured via an online survey (University of North Carolina at Chapel Hill Institutional Review Board 15-0379). The respondents, recruited individually, included twenty-four early career researchers (graduate students and postdoctoral associates) and five faculty.

Respondents were presented with one of eight faux schedules. Each schedule consisted of two timeslots. First, two timeslots each were randomly selected from the automated schedule and the manually modified schedule. These were then combined to produce all eight possible schedules consisting of one timeslot from the automated schedule and one from the modified schedule (Fig. 4). Each timeslot contained 14 concurrent sessions, and each session had a maximum of five talks. Each respondent was randomly assigned one of the faux conference schedules and a corresponding book of abstracts. Testing was blind in the sense that respondents were aware of the purpose of the study but not of which timeslot originated from which source (automated or manual).

Figure 4 Generation of faux schedules for human evaluation.

(A) The original automated and manually modified schedules are depicted here for purposes of illustration with six timeslots each. (B) Two timeslots from each schedule in (A) are randomly selected. (C) The eight possible schedules consisting of one each of the automated and modified timeslots.

The survey contained two groups of questions. First, we asked respondents to select the five talks they would like to attend within each timeslot, irregardless of whether they were assigned to the same session. We could then compare the automated or modified timeslots with respect to how the selected talk pairs were grouped into common sessions.

Secondly, we asked respondents to choose one session to attend in its entirety in each timeslot and report on the difficulty of finding a session where all the talks interested them. Responses were scored on a Likert scale of one to five with one being “very difficult”, and five being “very easy”. These responses could then be used to compare the topical coherence of the sessions from the automated and modified schedules.

If either of the schedules (automated or modified) were more effective than the other at capturing the topics of relevance to our sample of mock conference attendees, we would expect to see respondents (a) select more talks in the same session(s) and (b) select higher values on the Likert scale for timeslots from that schedule. With respect to (a), we found no significant difference in the number of same-session talk pairs between the automated and manual timeslots (unpaired t-test t =  − 0.720, p = 0.474, n = 29). With respect to (b), the responses for the automated and manually modified timeslots were quite similar in distribution (Fig. 5). The mode for the automated timeslots was four while that for the modified timeslots was three. Two respondents rated the selection “very easy” for the modified timeslot while none did for the automated one. While the expert evaluation does not reveal substantial differences between the automated and manually modified schedule in terms of preference by the survey takers, the limited size of the survey should be noted.

Figure 5 Responses of mock conference attendees when asked to rate the ease or difficulty of selecting a single session when presented with faux schedules containing one timeslot each from the automated and manually modified Evolution 2014 schedules.

The scale ranges from one (very difficult) to five (very easy).

Discussion

Manual scheduling of conferences is complicated, time intensive, and may often result in a suboptimal schedule with sessions that could be more topically coherent and timeslots in which sessions could be more topically disparate. Here, we have proposed and tested a strategy for automating the conference scheduling problem. In our approach, we first use topic modeling to identify latent topics and use the resulting weight vectors to measure similarity among talks and sessions. Stochastic optimization is then used to generate schedules according to the discrimination ratio, which simultaneously quantifies within-session coherence and between-session disparity.

In a comparison of different approaches for generating starting schedules and improving upon them, we found that Integer Linear Programming produced the best starting schedule, but that further stochastic optimization greatly improved upon the solution found by ILP. We attribute the inability of ILP to maximize the discrimination ratio to the heuristic compromise of splitting the problem into smaller sub-problems, which was necessitated by the size of the real-world problem instances. We also found that the initial schedule had little to no effect on the discrimination ratio of the final schedule. Thus, we recommend using a random or greedy algorithm to generate the starting schedule, since these approaches are less computationally expensive and easier to implement.

We found that Simulated Annealing performed better than naive Hill Climbing as a stochastic optimization strategy. If the results we obtained for the Ecology 2013 dataset are representative, and we accept the discrimination ratio is a reasonable objective function, then it appears that manually generated schedules can be far from optimal. This could be due to a number of reasons, apart from the obvious explanation that the combinatorial space of possible schedules is too large for humans to effectively search and evaluate. We cannot exclude that human conference organizers weigh additional factors (e.g., aiming for presenters within a session to represent a mix of different career stages). We would expect some difference between human perception of talk similarity and the inference of the same based on a topic model. And we would also expect a difference in how humans weigh coherence within sessions and disparity between sessions.

In fact, we did receive feedback from Evolution 2014 organizers that we should consider coherence first and disparity second. However, we saw that schedules produced in this way were inferior as judged by the discrimination ratio, although we do not know if they would be judged inferior by humans. This might be due to the way the algorithm operates—optimizing coherence within sessions first without regard to disparity between concurrent sessions. Once the coherence with sessions has been optimized, the algorithm is not allowed to change the placement of talks in sessions to maximize disparity but can only change the placement of the concurrent sessions with respect to each other. This results in a smaller search space for increasing disparity between concurrent sessions which might lead to lower D scores for schedules produced using these approaches.

Scheduling constraints are a regular feature of conferences, and initially we anticipated that they would be more troublesome than they ultimately proved to be. We found no decrease in discrimination ratio when incorporating constraints in the Evolution 2014 schedule. We hypothesize that the applied scheduling constraints were not restrictive enough to substantially limit the search space. For context, while approximately 24% of the talks had scheduling constraints, the majority could still be placed in 91% of sessions. In cases where constraints are more restrictive, one could modify the approach here to accept solutions that minimize the number of constraints violated, or weight the constraints such that solutions aim to minimize the total weight of violated constraints.

With the Evolution 2014 schedule, we took advantage of the opportunity to conduct a preliminary investigation into how much session-hopping users felt would be necessary in the automated schedule versus the manually modified one. By the two measures we looked at, prospective conference goers with expertise in the field found the two schedules to be comparable. Given the substantial changes made to the automated schedule, it was perhaps surprising that results did not show greater differences.

One possible interpretation of this result is that while the conference organizers may have modified the schedule in an effort to optimize their own subjective similarity and disparity measures, they did not improve upon the automated schedule from the perspective of a community of conference attendees with diverse interests. This also suggests that it would be reasonable for future conference organizers to use an automated schedule as is, without expending additional human effort vainly trying to improve upon it. However, a number of limitations with this experiment should be noted. The sample size was small and a limited array of schedules were presented for evaluation. While all survey participants had expertise in some area of evolutionary biology, we might have been asking them to evaluate sessions outside of their specific interests. And they were tested in an artificial setting; their behavior in navigating a real conference schedule may differ.

Taken together, we believe this work makes a number of contributions. First, topic modeling provides a reasonable input for automated clustering of conference abstracts. The scalability of this approach is attractive for large conferences. Secondly, D is a reasonable objective function, though a difficult one for humans to manually optimize. It’s value lies in capturing both similarity within and dissimilarity between sessions, the latter of which has been previously neglected. Third, we have identified fast heuristics for optimizing D.

Future Work

Would it be possible to improve upon this approach such that an automated schedule would be preferred by a future organizing committee to a manually generated, or manually modified, schedule? One area for potential improvement would be to customize the weights given to topics based on the perceived importance of conference attendees. In the approach described here, each topic received equal weight. However, a community of scientists may consider some topics more important than others. Values for the weights could be gathered by means of a survey or other user-contributed data. If topics were mapped to an ontology, weights related to the information content of topics could provide an indirect measure of importance (Resnik, 1995) without the need for a survey.

Given the comparable performance of the automated and manually modified Evolution 2014 schedules, it would be of interest to further examine how well statistical measures of topic similarity between talks match human perception. For similarity measures that do match well, it would then be of interest to see how sensitive humans are to schedules with very different levels of D, or to independently varying levels of similarity within sessions and dissimilarity between sessions.

Another pair of factors not considered was co-author and co-citation networks. Intuitively, talks that are closely linked in either kind of network may be similar in ways that are somewhat independent of how they are related by the topic model (Yan & Ding, 2012). Use of such network information could also help ensure that talks by individuals with strong intellectual ties are assigned to the same session or at least not assigned to different concurrent sessions.

Our implementation limits concurrent sessions to those that overlap fully. Conferences sometimes schedule sessions of differing lengths that partially overlap with one another, and accommodating this in future versions could allow for greater flexibility.

The heuristic approaches presented here have not been evaluated with respect to an exact approach with an optimality guarantee. Future work may consider developing exact approaches, such as Mixed-Integer Linear Programming, to better understand the computational bounds of these approaches and investigate if the heuristics proposed here are substantially faster as compared to exact approaches and if the solutions are comparable.

Conclusions

Automated scheduling of large conferences is a problem of great interest and utility to scientists across various domains. Here, we presented heuristic algorithms for the creation and optimization of conference schedules with concurrent sessions based on an objective function. The methods presented here are capable of “reading” conference talks, assessing similarities between the talks, and using those similarities to populate conference sessions. While these methods are a step forward in the field of automated conference scheduling, further work is needed to develop objective functions that accurately reflect user perception of “good” conference schedules.

Data and Software Availability

Data and software for this work are available at https://doi.org/10.5281/zenodo.2860367.

Supplemental Information

Appendix S1 Samples of Automated, Random, and Manual conference schedules and the experiment survey used in the study

Click here for additional data file.

We wish to thank J Scott Provan for his guidance on implementing the ILP schedule creation approach. We thank the Evolution 2014 program committee (J Cryan, E Lacey, K Pfennig, B Langerhans, C McClain, B O’Meara, A Rodrigo, M Servedio, J Shaw and J Thorne) for their time and expertise in preparing the final Evolution schedule from our automated preliminary schedule. We thank our survey participants who enabled us to enable comparisons of the automated and manual schedules. Finally, we extend our thanks to Christian Santiago, Daphne Klotsa, David Egolf, Itai Cohen, John Crocker, Karen Daniels, Michelle Driscoll, and Peter Olmsted from the American Physical Society for providing pictures of preparing the schedule for their 2017 meeting.

Additional Information and Declarations

Competing Interests

Author Contributions

Ethics

Data Availability

Todd J. Vision is an Academic Editor for PeerJ. Katherine Lamm is employed by ROI Revolution, Inc.

Prashanti Manda conceived and designed the experiments, performed the experiments, analyzed the data, contributed reagents/materials/analysis tools, prepared figures and/or tables, performed the computation work, authored or reviewed drafts of the paper, approved the final draft.

Alexander Hahn performed the experiments, analyzed the data, contributed reagents/materials/analysis tools, prepared figures and/or tables, performed the computation work, approved the final draft.

Katherine Beekman conceived and designed the experiments, performed the experiments, analyzed the data, contributed reagents/materials/analysis tools, performed the computation work, approved the final draft.

Todd J. Vision conceived and designed the experiments, contributed reagents/materials/analysis tools, prepared figures and/or tables, authored or reviewed drafts of the paper, approved the final draft.

The following information was supplied relating to ethical approvals (i.e., approving body and any reference numbers):

The University of North Carolina at Chapel Hill granted IRB approval for a survey conducted as part of this study (IRB 15-0379).

The following information was supplied regarding data availability:

Data and software for this work are available at Zenodo: Manda, Prashanti, Hahn, Alexander, Lamm, Katherine, & Vision, Todd. (2019, May 16). Avoiding ”conflicts of interest”: A computational approach to scheduling parallel conference tracks and its human evaluation - Data and Software. Zenodo. http://doi.org/10.5281/zenodo.2860367.

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
