# Peer review of "Avoiding “conflicts of interest”: a computational approach to scheduling parallel conference tracks and its human evaluation"

_PeerJ Computer Science, doi:10.7717/peerj-cs.234_

## Round 0.1 · original submission · Major Revisions

The reviewers enjoyed many aspects of the paper, and appreciated the writing and the provision of data. However, they have identified serious flaws with the experimental design and additional flaws with the validity of the findings. One reviewer felt the review to be out of scope. The reviewers were no clear on the novel contributions of your approach, and if this is a case report on existing methods rather than a research article, then this is out of scope.

If you resubmit, then you should address the comments of all reviewers. In particular:

You should cite appropriate literature in the field, and clearly address which novel contributions your approach makes, in terms of theory and performance. It should be clear if your method is entirely automatic.

Case studies are not in scope for PeerJ. If you resubmit, then the resubmission must be clearly a research article.

For the evaluation of the results, you should use all appropriate statistical tests and present appropriately. When comparing with existing methods, you should be clear that you are comparing like with like, e.g. if your method involves manual intervention, then include a comparison with another method that also involves intervention.

Reviewer 1 ·

Basic reporting

See PDF.

Experimental design

See PDF.

Validity of the findings

See PDF.

Additional comments

See PDF.

Annotated reviews are not available for download in order to protect the identity of reviewers who chose to remain anonymous.

Reviewer 2 ·

Basic reporting

Writing is clear and unambiguous, with professional word choice throughout.
Article structure and general presentation is high-quality. Spacing of equations and tables could be improved (there is too much space), but this seems like a pretty minor quibble.
Could be useful to mention time-frame of talk scheduling; shorter turnaround time helps motivate the need for an automated method.
Citations are improperly embedded in sentences in some places (e.g. line 185, 190, etc.)
Data is made available online.
Related work is a little sparse, but it could be because (as the authors point out) there may not be much related work on conference scheduling.
Topic modelling is typically introduced with the Blei 2003 citation.
Text should stay within boundaries of image in Figure 3, “g” in “algorithm” is cut off in Figure 3.

Experimental design

Contributions are:
Use of topic modeling to measure talk similarity combined with stochastic optimization of a global objective that targets both intra-session similarity and inter-session dissimilarity
Lack of a need for human intervention
The lack of a need for human intervention seems to be contradicted by the fact that one of their test cases required significant modification before being deployed in the real world
On line 300, it’s said that 0.5% of talk pairs that share a session in the automated schedule are retained in the modified schedule for Evolution 2014
None of the experiments indicate that the automated method is either necessary or sufficient for producing reasonable conference schedules, especially in light of the extreme changes required by the conference organizers. 2 potential suggestions to loosely show necessity and/or sufficiency of the automated method are:
An analysis of the differences between the manual and the automatic schedules. The authors could provide an appendix with some sample time slots from the manual and automatic schedules. Note that in the manual schedule (https://www.evolutionmeetings.org/uploads/4/8/8/0/48804503/2014_program.pdf), sessions are labelled with easily understood subject areas. Can the automated sessions be labelled in this fashion? One or two examples would go a long way. It would also be nice to see what a time slot would look like if all sessions were created at random, or using the greedy initialization method.
Adding another automated method to the user survey to serve as a baseline. It is unclear whether the similarities between automated and manual schedules in terms of the 2 metrics derive from success of the automated method or from wording of the survey questions and/or desire of the annotators to “settle” for reasonable schedules when there exist much better schedules. That is, it is not apparent that a very simple baseline (or a random approach) would not satisfy the 2 criteria assessed in the survey.
Many of their methods were straightforward applications of existing methods, and those were described adequately. Some parts may not have been described enough to enable full replication (e.g. line 192, where they describe their use of “a modification of Kruskal’s algorithm”), but overall it is well done.
A survey is certainly the right kind of evaluation for this problem. This kind of thing really needs human evaluation.

Validity of the findings

Findings reported in the abstract are misleading - For example, the abstract claims that “[the authors] find that simulated annealing improves the objective function over an order of magnitude relative to the schedule manually produced by the program committee,” yet it is not clear here that the objective function is a newly defined objective that does not have proven relevance to conference scheduling yet (objective is motivated by the Evolution 2013 blog post, but is not directly shown to be relevant to the assessed quality of resulting schedules).
Conclusions are well-stated, with some (but not a lot) of discussion of further improvements. The suggestion to incorporate co-authorship and co-citation networks seems like a good one.
It is interesting that scheduling constraints did not significantly impact the discrimination ratio, and that Evolution 2014 organizers suggested focusing on coherence within sessions rather than disparity between sessions.
No discussion of why the HC-A and SA-A strongly outperform the HC-B and SA-B methods, which would be interesting, though not necessary.

Additional comments

I find the scientific results weak: both a straightforward application of previous methodology, and somewhat misleading statements of conclusions from results.

The paper is, however, well-written, and it’s nice that the method was tested on real conferences. The authors describe their approach well enough. One particularly nice touch was the inclusion of images of manual conference organization in Figure 1, which gives the reader a strong understanding of the difficulty of manually scheduling conferences.
There are some minor formatting issues that could be improved, particularly with regard to the amount of whitespace and the formatting of equations (for example, equations should be centered).
I appreciate that the data is made available online.

Reviewer 3 ·

Basic reporting

Clear and unambiguous professional English is used throughout.

Although unusual to include photos like those in Figure 1 in technical papers, they perfectly illustrate the arcane current manual practices that are still so widespread.

123. An introductory sentence to set out the structure of the METHODS section would let the reader know what is coming in sections 1.1 to 1.6.

155. Cosine similarity will be between 0 and 1 where the vector elements contain only positive values. Is this is the case here, rather than -1..1?

157. "We introduce an objective function called the Discrimination Ratio...". Is this ratio new and something introduced here? It reminds me of cluster analysis measures and the case for novelty would benefit from some justification as to why the Discrimination Ratio is different to some of the internal measures often used to evaluate cluster quality. Highlighting the difference would substantiate the claim. The novelty is most likely to be in the use case, in which case I'd suggest including a reference to a standard cluster analysis function.

164. A sentence spelling out when the ratio will be high, middle and low will help the reader to remember the intuition.

203. "T_{m}ax" needs explaining or should this be T_{max}?

217. "a Greedy or [an] ILP schedule"

240. "typically have [constraints?] that restrict"

241. "talks competing [in?] awards"

264. "deemed" has negative connotations and "judged" would be a better word to use here

265. You should briefly mention how you get from topics to pairwise similarity (weight vectors?)

296. "following manual procedures that elude easy description", made me laugh out loud, but it is true and underscores how arbitrary assignment is.

300. Given the substantial changes made by the program committee (affecting 99.50% of pairs), some further details about the changes would help the reader to understand the result. For example, it might be interesting to compare the before/after Discrimination Ratios of the two assignments (and the two components of the ratio) and/or topic extraction (or top tfidf terms) of the resultant sessions to see whether there is a pattern to the adjustments. This may go some way to investigating whether the later feedback from the committee about "coherence first and disparity second" is supported/evident here.

303. As it is, this major disagreement between the assignments is left hanging in this very short paragraph and the paper launches straight into the much longer 2.2.2 without really finishing off 2.2.1. This is a shame because 2.2.1 should not be seen as a negative result and that point should be made before going on to almost defend the result in 2.2.2. There is no objectively correct schedule, a point made later in the discussion, but perhaps trail that idea here and be less defensive of the differences; instead focus on what can be asked and learnt from the difference.

Table 2 could be made more readable by reducing the width of the left column and widening the middle.

Table 3 could be made more readable by widening to leftmost column to avoid word wrap and, perhaps, right aligning the value columns.

Tables 1 & 4 are overly wide for their content, making them harder to scan from variable to variable description. Suggest reducing overall width or reducing left column width.

Section 3, DISCUSSION, would benefit from splitting in two so that that concluding comments and future work is presented under its own heading. The discussion could then focus on discussion of the results presented here.

Experimental design

There are a lot of uncontrollable externalities when experimenting on real-world conferences, but the authors have done a good job in designing experiments within these constraints. The experimental design for automated scheduling using 2013 conference data to inform their experiments on the core problem of the 2014 conference is principled and well thought through.

While the expert analysis (2.2.2) to compare automated and manual schedules is well designed, small number statistics dictate that no strong conclusions can be drawn from 29 data points. The work is still interesting and thought provoking, but it should be presented as a preliminary study to suggest further work rather than anything to draw firm conclusions from. It is still important to publish and share such work, but its presentation needs to clearly state that this is only an exploration rather than a full rigorous experiment.

I commend the authors for observing best practice open science and publishing their software and data on Zenodo. A few comments below:

- README.md does not state python version and other dependencies, doesn't include instructions for running the programs or how to reproduce the results in the paper. An orientation of the file tree in the readme would be useful given there are so many folders. Minor point: commentary refers to 2014 in future tense.

- Readability of the python code would benefit from pylint/pep8 formatting (which can be done automatically in many editors, so not a good excuse for not following best practice).

- A comment at the head of each file would make the code easier to understand.

- The inclusion of import statements (rather than just the call to main()) is a little confusing and can upset code assist/linting in some editors; even if the code is invoked via inclusion rather than as main, the imports will only happen once no matter how many times they are invoked.

- Minor tidying up suggestion: remove the Mac ".DS_Store" files scattered throughout the tree.

Validity of the findings

By using post-hoc analysis of 2013 conference data to inform the live scheduling for a much larger 2014 conference timetable. The smaller size of the problem addressed in the 2013 data enabled them to test multiple starting conditions that, by intuition alone, one might have expected to produce a better final allocation, as measured by the Discrimination Ratio. The result that the starting condition was largely unimportant was surprising and is potentially a valuable insight into the dominant constraints in the scheduling problem - irrespective of the subsequent major adjustments made manually by the program committee.

One criticism of this and other work in this area is the lack of longitudinal, historical accumulation of data from past conferences to feed into the current scheduling. It is common to use authors' past publications to profile their suitability for reviewing papers submitted for peer review but I have not yet seen the same historical perspective applied to session scheduling. True, the paper makes a (probably correct based on personal experience) claim that schedule conflicts are a common feedback topic in conference feedback. Future work might perhaps make such feedback one of the inputs to scheduling but before that can happen, the community needs better tools.

Additional comments

Not scientifically important, but readers may be curious as to why results from 2013-14 conferences are being reported now. This is not a criticism but perhaps some narrative could be given to say why now. My own thoughts are that mainstream practice and tools for conference organisation have hardly changed since 2013-14 and there is little re-use of prior work where organisers decide to build conference support tools, so publishing this work and supporting codes now is a valuable and welcome contribution to the field.

---

## Round 0.2 · Minor Revisions

Reviewer 1 suggests minor changes in basic reporting which should be implemented.

This reviewer also argues that the results are too weak to warrant publication. However, this journal does not make decisions based on perceived impact, degree of advance or novelty, so do not require additional experiments as suggested. Nevertheless, the suggestions could also be incorporated in the discussion/future work section.

The research should be valid and justified. Reviewer 1 provides 4 cases where statements in the paper are not justified by the results. These statements should either be weakened/qualified such that they are valid, or sufficient justification as stated by the reviewer must be demonstrated. All 4 of these must be addressed.

Reviewer 1 ·

Basic reporting

- Although I still doubt the added value of Figure 1, I respect the authors' decision to keep it in the manuscript.
- Compared to the previous version, the literature section is now more comprehensive.
- Line 122: the IP abbreviation has not been used/explained before.
- Line 130: briefly explain how attendance is measured.
- Line 147: I would replace "the solution aims to" with "the goal is to".
- Line 158: the word 'profiles', i.e. binary participant preference vectors, is not explained and can be confusing to the reader.
- Line 305: I would rephrase “the ability to escape” to e.g. “potential/possibility to escape”.
- I appreciate the inclusion of a "Preliminaries" section. However, the caption of Table 1, as indicated in my original comment, is still false, as the table does not contain a single variable. Rather, the table exclusively contains parameters. This has an impact on line 208 as well. Regarding the SA entries in Table 1: the temperature Z is included in Table 1, yet the constant alpha is not. The Y entry is coupled with the description "number of seed talks for the greedy algorithm". The word algorithm should be changed to heuristic. Same for the R-entry.
- I would suggest a more precise title for Section 1.7: Discrimination ratio metaheuristics
- The legend for Figure 3 is awkwardly placed. Why not, for sake of consistency, do the same as in Figure 2?
- Line 487: typo 'althogh'.

Experimental design

- I am still of the opinion that the scientific results are too weak/insufficient to warrant publication. A number of my previous suggestions for improving the paper were relegated to 'future research'. In what follows, I provide 3 concrete suggestions which, when properly implemented and included in the paper, can make the paper salvageable. I am sympathetic to the authors and know these changes will take significant time/effort, but, when made, the quality and contribution of the paper would be greatly improved.
Suggestion 1: provide an exact approach (a MIP-formulation), along with bounds (can just use the current reported best bound is the computation times are extensive, e.g. > 2 hours) and computational results. The heuristics would be better motivated, if it turns out they are (i) much faster than the exact IP approach and/or (ii) give relatively good solutions. The current motivation in lines 273-274 is simply insufficient.
Suggestion 2: in Section 1.6, provide ‘sufficiently precise’ pseudo-code for the generation of initial schedules (i.e. separate pseudo-code for 1.6.1, 1.6.2 and 1.6.3), so that they can be independently implemented and lead to similar results. In that pseudo-code, also include how you deal with the constraints in L, as well as the how many schedules are generated. In addition, the authors should mention whether it is checked that all generated initial schedules are distinct. Are duplicates possible?
Pseudo-code is also required in Section 1.7.
Suggestion 3: also add computation times for the metaheuristics. Compare these with the exact approach. Also compare the solution values with either exact solutions or some bound.
With these suggestions, the paper can be a proper benchmark for future studies. Additionally, the contribution w.r.t. computational performance would clearly make the article in-scope for PeerJ CS.

- Lines 262-265: are the random schedules all unique?
- Line 268: how is Y changed for the R different schedules? And are those initial schedules unique?
- Line 307-308: is it checked that all initial schedules are unique?
- Line 316: how were Z and alpha set?
- Line 344: the metaheuristics are not guaranteed to find even a feasible solution. A few sentences explaining what can be done in such a case would be in place. Some options I can think of: separate constraints in hard and soft constraints, or minimize the number of violated constraints. Taking this into account in the current metaheuristics is not required.
- Line 362: So R = 50?
- As the authors indicate in their reply, the bars representing two standard errors in Figures 2 and 3 are barely visible. I completely missed them in my original review. My suggestion is then to make those bars more visible (e.g. by enlarging them).

Validity of the findings

- Lines 273-274: this statement is not backed up by findings at all. Some IP models with millions of variables can be solved in mere seconds. The authors can, as mentioned before, implement an exact approach and provide computational results to back up this claim.
- Lines 394-397: without an exact, i.e. optimal, solution, the authors can't really make interesting statement regarding the presence of constraints.
- Lines 461-462: this could still be the case after the metaheuristics in this paper are applied.
- Lines 519-520: the authors claim to present fast heuristics. However, this claim is unsubstantiated, as not a single computation time is given in the paper (see suggestion 3 as well).

Additional comments

- The authors have clarified the concepts 'session' and 'concurrent'. However, in line 182, I think it would be better to state the starting and ending time of concurrent sessions coincide/are the same, rather than just overlap. Overlap might mean partial overlap, which is not allowed. This would be more in line with lines 540-542.
- My original comment regarding algorithms and heuristics is still not fully addressed. In Sections 1.6 and 1.7, the (false) usage of the word algorithm is still rampant. It also occurs in Section 2.

---

## Round 0.3 · Minor Revisions

I would strongly recommend you follow the standard practice of presenting the algorithms as pseudocode. Even though you have narrative descriptions of these, and open reference implementations, pseudocode is standard for CS papers describing algorithms. Although the overall length of the paper will increase, it will be easier to understand precisely what each algorithm is doing.

You may also wish to carry out the additional experiments as suggested by the reviewer, but this is not required. I recommend noting the possibility of doing this in the discussion/future work section

---

## Round 0.4 · accepted · Accept

I thank you for your attention to the comments, and for adding the pseudocode sections.